# Network analysis of physical activity and depressive and affective symptoms during COVID-19 home confinement

José A. Cecchini, Alejandro Carriedo , Antonio Méndez-Giménez and Javier Fernández-Río

EDAFIDES Research Group (Education, Physical Activity, Sport, and health), Department of Education Sciences, University of Oviedo, C/ Aniceto Sela, s/n, Oviedo, Spain

## Research Article

quarantine; coronavirus; physical activity psychological effects; mental well-being

**Corresponding author:**
Alejandro Carriedo;
Email: carriedoalejandro@uniovi.es

## Abstract

Introduction: The aim of this study was to analyze the network structure of physical activity, frequency, depressive, and affective symptoms in people under home isolation due to COVID-19. Method: A longitudinal study was conducted in two phases (beginning (March 19, 2020) and end of home confinement (April 8, 2020)). The sample consisted of 579 participants from Spain (250 men and 329 women) aged 16 to 92 years (overall sample: $M = 47.06$, $SD = 14.52$). A network analysis was performed. Results: Four clusters emerged (PA, depressive symptoms, positive affect, and negative affect). A higher frequency of physical activity was related to better-sustained attention, increased alertness, and enthusiasm. In addition, feelings of guilt and shame were mitigated, and confinement distress and irritability were reduced. Physical activity also mitigated fatigue in women, whereas feelings of unhappiness were reduced in men. Conclusion: Physical activity seems to be an effective option for mitigating the negative effects of the COVID-19 pandemic. Public health policymakers should develop programs to promote physical activity in order to be able to cope with confinement or similar scenarios in the future.

## Impact statement

- This study examined the associations between physical activity, depression, positive affect, and negative affect through a network approach.
- This is a longitudinal study that shows the relevance of higher frequencies of physical activity for mental health in confined population.
- This is the first study that has adopted a network approach to examine these variables.

## Introduction

The United Nations health agency (UN, 2022 has warned about an increase in deaths in most regions of the world due to COVID-19. Thus, all available tools, not just vaccines, should be used to combat the coronavirus. More than two years after social distancing measures were first implemented, Shanghai returned to forced confinement on March 28, 2022, due to an increase in COVID-19 cases. Several lockdown measures have been established in some regions of China (e.g., Xinjiang had been under lockdown for more than 100 days), indicating that the Omicron subvariant may still threaten to lead to more frequent lockdowns (Yeung, and CNN's Beijing Bureau, 2022).

### *Confinement and psychological health*

On March 14, 2020, Spain declared a state of alarm that imposed strict isolation for the entire population. Freedom of movement was limited to basic activities such as buying food or going to health centers. This situation lasted for 50 days. During the first week of confinement, levels of physical activity (PA) fell by 20%, which led to only 48.9% of the population meeting the PA levels recommended by the World Health Organization (López-Bueno et al., 2020). Other studies have also observed the same result (Sebastiano et al., 2020). COVID-19 home isolation also affected the weight of the Spanish population, expanding depression and obesity, which was related to a decrease in PA (Fernández-Río et al., 2020). Depressive symptoms are endorsed to daily life stress (e.g., family, work, school, and friends), but, in isolation contexts, stressors include direct causes (e.g., fear and uncertainty). Affect refers to a mental state that includes feelings of self-evaluation (e.g., happy–sad and good–bad). However, positive affect indicates the degree to which a person feels enthusiastic, while negative affect reflects the degree of

subjective distress (Watson et al., 1988). Positive affect has been linked to psychological well-being, while negative affect has been connected to destructive mood states (Watson et al., 1988). Stressful scenarios tend to produce an increase in negative affect, making individuals more exposed to develop psychological problems such as depression (van Winkel et al., 2015). Positive affect has been positively related to physical activity, while lower negative affect was associated with higher physical activity (Niermann et al., 2016). Greater positive affect and lower negative affect were also related to PA during the COVID-19 pandemic (Carriedo et al., 2020b). During this period, there was also a significant increase in depressive symptoms (Cecchini et al., 2021). The study found that a moderate intensity of PA could have been sufficient to prevent an increase in depressive symptoms during home isolation. PA intensity can be classified by rate of energy expenditure (i.e., metabolic equivalent of task (MET)). It is commonly established that light PA (e.g., casual walking and stretching) requires less than 3 METs, moderate PA (e.g., playground games and dancing) requires between 3 and 5.9 METs, and vigorous PA (e.g., soccer and swimming) requires more than 6 METs (Ainsworth et al., 2011).

The relationship between PA and psychological health has been analyzed during the pandemic. PA may have an important role in reducing the consequences of the COVID-19 pandemic (Sallis and Pratt, 2020), for example, by strengthening the immune system and reducing inflammation (Hojman, 2017). PA has also been associated with mental health and sleep quality (Firth et al., 2019). For example, higher levels of PA are associated with lower symptoms of depression and anxiety (Chekroud et al., 2018). In this context, it was observed that those individuals who got involved in more moderate–vigorous PA showed fewer symptoms of anxiety and depression (Barcelos-Mendes et al., 2021). Méndez-Giménez et al. (2021) also discovered that volume (METs-min/week), frequency (days/week), and volume (hours) of PA were inversely associated with the likelihood of developing depressive symptoms during home confinement. Other studies that have related PA to resilience during the nationwide lockdown in Spain concluded that individuals who regularly participated in vigorous PA during the first week of confinement reported higher resilience in terms of higher scores in locus of control, self-efficacy, and optimism (Carriedo et al., 2020a). Resilience and PA may act as protective elements in stressful situations such as home confinement in Spain (Fernandez-Rio et al., 2022).

### Methodologies for COVID-19 research and data analysis

Traditional approaches have investigated psychological disorders by adding together the different symptoms of a specific disorder and using a total score as a measure of the so-called latent variable or factor (e.g., depression; Borsboom, 2008; Schmittmann et al., 2013). However, not all people with the same diagnosis experience the same symptoms. For instance, two people might be diagnosed with the same disorder with only a single symptom in common. In addition, many studies have found that individual symptoms of a particular psychiatric diagnosis may be related to different risk factors or causes (Fried, 2015; Mamun et al., 2020; Wasserman et al., 2021).

Recently, a new approach to understanding the complex relationships between psychiatric symptoms has emerged (Fisher et al., 2017), called network analysis. This approach is based on the premise that the different symptoms interact with each other

(Jones et al., 2021), so it is necessary to understand the strength and nature of these interactions (Beard et al., 2016). In network analysis, nodes represent psychiatric symptoms and edges between nodes reflect the relationships between symptoms, including the activation of one symptom to other symptoms across the network (Borsboom, 2017; Rouquette et al., 2018). Consequently, network analysis serves to explain patterns of connection between individual psychiatric symptoms and also between psychiatric disorders (Borsboom and Cramer, 2013). It is also necessary to understand the affective behavioral and cognitive mechanisms that contribute to the maintenance of psychiatric symptomatology, as they are predisposed to be manipulated by different pathways (e.g., PA).

One of the main strengths of network approaches is their ability to analyze the dynamic relationships between psychological constructs and other variables, such as PA. These relationships simultaneously show which network connections are shared among participants, which contributes to the generalizability of the findings (Hofmann and Curtiss, 2018). Another strength of the network approach is that complex multivariate models can be visualized in an insightful and illustrative manner (Bringmann and Eronen, 2018). Likewise, visual inspection of a network can convey relevant information with minimal effort (Costantini et al., 2019). Network approaches are also valuable in providing an alternative psychometric operationalization of the affective dynamics underlying PA (Curtiss et al., 2019). However, an important limitation of this network approach is that the methods it employs can lead to model overfitting. Model overfitting refers to when it becomes more complicated than necessary and involves modeling spurious relationships (Beltz et al., 2016). This has called into question the replicability of networks derived from sample data (Forbes et al., 2017; Fried and Cramer, 2017). For this reason, it has been recommended that, although network approaches are a useful exploratory tool, results derived from these models should be followed up with further confirmatory research (e.g., cross-validation or experimentation; Epskamp et al., 2018).

### Statement of the study

Different studies using network analysis have examined depression (Bai et al., 2021; Di Blasi et al., 2021; Ebrahimi et al., 2021; Liu et al., 2021), anxiety (Bai et al., 2021; Heeren et al., 2021), and emotions (Martín-Brufau et al., 2020) during the COVID-19 pandemic. As aforementioned, other studies have analyzed the relationships between PA and different psychological factors during home confinement (e.g., López-Bueno et al., 2020; Carriedo et al., 2020a, 2020b; Cecchini et al., 2021; Méndez-Giménez et al., 2021 and Fernandez-Rio et al., 2022). However, these studies used other approaches such as generalized linear models (Carriedo et al., 2020a), analysis of variance (López-Bueno et al., 2020; Carriedo et al., 2020b), linear mixed models, multilevel modeling (Cecchini et al., 2021), latent class analysis (Fernandez-Rio et al. (2022), and logistic regressions with restricted cubic splines (Méndez-Giménez et al., 2021).

Hence, there is still much to learn regarding the relationship between PA and psychological outcomes. To our knowledge, no study so far has used network analysis to explain the relationship of PA with depressive symptoms and positive and negative affects during home isolation. This new approach could lead to a better understanding of this line of research and help explain how these constructs and mechanisms are related (Marconcin et al., 2022).

For instance, could a network analysis explain patterns of connection between depressive symptoms, affect, and PA performed during the beginning and end of home confinement? How are the dynamic relationships between these variables? What network connections are shared among participants?

Based on this, this study aimed to conduct a network analysis to explore the relationships between weekly PA frequency (Méndez-Giménez et al., 2021), depressive symptoms, and positive and negative affects during home confinement in order to analyze whether individual symptoms may be related to different types of PA and the centrality of these symptoms (Fried, 2015; Mamun et al., 2020; Wasserman et al., 2021).

## Methods

### Participants

The sample consisted of 579 participants from all regions of Spain (250 men and 329 women) aged 16 to 92 years (overall sample: $M = 47.06$, $SD = 14.52$).

### Instruments

*The International Physical Activity Questionnaire (IPAQ)* is an instrument that was originally developed for cross-national monitoring of PA and inactivity (Craig et al., 2003). The IPAQ exhibits acceptable measurement properties, and it is as accurate as other established self-reports, having demonstrated reasonable measurement properties for monitoring levels of PA in different populations (Craig et al., 2003). This study used the Spanish adaption of the IPAQ short form (IPAQ-SF) "last 7 days recall." This version provides information about the days per week on which an individual participated in light, moderate, or vigorous PA for more than 10 minutes (days per week involved in light, moderate, or vigorous PA = regularity of PA throughout the week).

*Depressive Symptoms.* A Spanish version (e.g., Cecchini et al., 2021) of the six-item self-report scale developed by Kandel and Davies (1982) was used. Participants respond to questions beginning with the phrase: "During the past 12 months, how often…". The six questions are related to social activities that may affect their health. Participants indicate whether the situation occurred "often" (4), "sometimes" (3), "rarely" (2), or "never" (1). To address the goal of this study, this question root was modified to match the time of confinement: "During the previous week of isolation, how often…" for each question (e.g., "…have you felt too tired to do things?"). Previous studies have reported acceptable reliability for the scale in Spanish populations (Cecchini et al., 2021). In this study, Cronbach's alpha for the full scale was .83.

*The positive and negative affect schedule* (Watson et al., 1988). This scale has 20 items divided into two subscales: positive affect (10 items) and negative affect (10 items). Both subscales are scored using a five-point Likert scale from 1 (*not at all*) and 5 (*strongly*). Internal consistency ranged from .86 to .90 in positive affect and from .84 to .87 in negative affect (Watson et al., 1988). In this study, participants reported about their mood during the previous week.

### Procedure

Permission to conduct the study was obtained from the researchers' regional Research Ethics Committee (no. 2020.165).

The participants completed a questionnaire twice during confinement. The first wave of data collection was on March 19, 2020, and the second was on April 8, 2020. Participants were contacted by email, Facebook, WhatsApp, YouTube, Instagram, and Twitter via a non-probability snowball sampling strategy to invite them to complete an online questionnaire. This sampling strategy is frequently used to identify potential subjects in studies where participants are difficult to find, such as in the present case where the entire population was confined to their homes. The inclusion criteria were aged ≥16 years and participants had to declared to their willingness to answer a questionnaire and a subsequent follow-up. The exclusion criteria were not to answer the follow-up questionnaire. It took about ten minutes to complete the questionnaire at the beginning of the study and the follow-up. For the size of the universe of 47,100,396 people (Spanish population in the year 2020), considering 50% heterogeneity, 5% error margin, and 95% confidence level, the number of participants needed was 385. The first page of the questionnaire informed the participants that their responses would be anonymous and that they could stop responding to the questionnaire at any time since participation was voluntary. Before answering the questionnaire, participants had to accept a clause on the first screen giving their informed consent. In the first wave of data collection, 641 people participated (men = 279 and women = 362). All people who participated in the first wave of data were invited to complete the same online questionnaire at the end of the home confinement. However, 62 people decided to withdraw from the study and they were excluded from the subsequent analyses. Thus, in the second wave of data, 579 people completed the questionnaire (men = 250 and women = 329).

### Data analysis

First, descriptive analyses were performed on PA levels and mental health variables. Cronbach's alpha, composite reliability rh, and average variance extracted were used as measures of reliability and validity. Then, network analyses were performed using the following packages in RStudio 2022.02 software: *bootnet* (Epskamp et al., 2018), *qgraph* (Epskamp, 2012), *EstimateGroupNetwork* (Costantini et al., 2021), *ggplot2* (Wickham, 2016), *dplyr* (Wickham et al., 2021), *stringr* (Wickham, 2019), and *ggthemes* (Arnold and Arnold, 2015). In network language, symptoms are represented in nodes, while correlations between individual symptoms are reflected in edges (Beard et al., 2016; Wang et al., 2020). In this study, 29 nodes were included: six depressive symptoms, 10 positive affect symptoms, 10 negative affect symptoms, and three dimensions of PA frequency – light, moderate, and vigorous. Recently, Epskamp et al. (2017) pointed out that repeated-measures data can be used to separate networks representing between-subjects variability from networks representing within-subject variability. To analyze the between-subjects network, the mean of each participant's scores on each variable across measurement points was calculated, followed by the estimation of a network over person means (Fleeson, 2001; Shiffman et al., 2008). The within-subjects network was calculated by centering the data on the person (Epskamp et al., 2017). Thus, the mean of each variable for each participant was subtracted from the raw scores of each participant and then a network was estimated over the centered data. This method can be applied even in cases where only two repeated measures per participant are available (Epskamp et al., 2017). Since it is important to compare network estimates across different groups (Costantini et al., 2019), in the present study networks were

compared by participant gender. Once the networks were calculated, different tools or indices were used to summarize the patterns of relationships in the network.

First, a visual inspection of a network is always useful because it conveys relevant information with minimal effort (Cramer et al., 2012). The Fruchterman–Reingold algorithms (Fruchterman and Reingold, 1991) place nodes close to each other if they are highly connected or further apart if they are not. Next, an attempt was made to identify the centrality of the nodes, that is, to determine whether some nodes were more influential than others. However, the concept of centrality is multifold and each index reports a specific type of centrality. A node can be central because it has strong direct connections with many nodes (strength centrality; Barrat et al., 2004), and a node can also be central because both direct and indirect routes connecting it to other nodes are generally short (closeness centrality). Likewise, a node can be central because it is often located on the shortest path between two other nodes and therefore is important in the connection those nodes have between them (betweenness centrality). Networks consist of nodes and edges. The nodes represent the objects or variables under study. Edges represent the connections between the nodes. In this network analysis, nodes usually represent symptoms and edges represent associations between them. These symptoms are of individuals, so the network modes are type 1, in which a single set of actors and the relationships that link them are studied. However, the limits of the network are the criterion that determines the membership of the actors in the network, denoting the social closure of that network, in this case subjects confined by COVID-19 between the ages of 16 and 92.

## Results

### Descriptive, reliability, and validity analysis

Table 1 shows the mean and standard deviations (SD) for men, women, and the complete sample in the two measurement points. Several *t*-tests for independent samples by sex were performed, showing that at both T1 and T2 women scored significantly higher than men on all items that measure depressive symptoms, except for the item "tired," and in six items of ten of the negative affect questionnaire. In contrast, men scored significantly higher than women on positive affect in some items (see Table 1). Regarding PA levels, women showed significantly higher values on days of LPA and men showed higher values on days of VPA levels. Repeated-measures *t*-tests were also performed between T1 and T2 for the whole sample. The results showed a significant increase in four of the six depressive symptoms and the items pa4 "enthusiastic" and na5 "hostile." An increase was also observed between T1 and T2 in days of VPA and days of MPA.

Cronbach's alpha, composite reliability rh, and average variance extracted were used as measures of reliability and validity. In depressive symptoms, Cronbach's alpha was .84; composite reliability (rh_a) was 0.87; composite reliability (rh_c) was 0.89; and average variance extracted was 0.57. In positive affect, Cronbach's alpha was 0.89; composite reliability (rh_a) was 0.90; composite reliability (rh_c) was 0.91; and average variance extracted was 0.51. In negative affect, Cronbach's alpha was 0.85; composite reliability (rh_a) was 0.88; composite reliability (rh_c) was 0.88; and average variance extracted was 0.51.

### Inter-subject network analysis

The network shown in Figure 1 (top) indicates that the three dimensions of PA grouped together to form a first cluster. A second cluster was formed by the variables making up positive affect. A third cluster was formed, due to proximity, by depressive symptoms, and a fourth, relatively separate grouping, by the dimensions of negative affect. The four clusters were not completely separate, as they were connected with weaker links, except for the strong links between "nervous" (nervous) (an8m) and "hopeless" (D4m) and between "distressed" (distressed) (an1m) and "unhappy" (D3). The connections between the PA dimensions and the other clusters were weak. The three PA dimensions were positively connected with the variable "attentive" (ap10m). The frequency of light PA (DLPAm) was furthermore negatively connected with "embarrassed" (ashamed) (an7m) and "guilty" (guilty) (an3m) and positively connected with "alert" (alert) (ap6m). DMPAm was negatively related to distressed (an1m), and DVPAm was negatively related to irritable (an2m) and positively related to enthusiastic (ap4m).

Figure 1 (middle and bottom) shows the results in the inter-subject network by participant gender. The four previously noted clusters that emerged in the full sample were present for both men and women. The inter-cluster links were weaker, but the links between "nervous" (an8m) and "hopeless" (D4m) were stronger for women than men and between "anguish" (an1m) and "unhappiness" (D3) were the same for both genders. The three dimensions of PA frequency were positively connected with the symptom "attentive" (ap10m), except for DLPA in women. The frequency of light PA (DLPA) was still negatively connected with "ashamed" (an7m) and "guilty" (an3m) for women, but only with "guilty" (an3m) for men. For women, there was a new negative connection between DLPA and "tired" (D1m). The negative relationship between DMPAm and "anxious" (an1m) remained only in women, and there was a new negative connection between DMPAm and "nervous" (an8m) for women only. Interestingly, for the men there was a new positive connection between DMPAm and "fearful" (an10). The positive relationship between DVPAm and "enthusiastic" (ap4m) only remained for women, and there was a new positive connection between DVPAm and "interested" (ap1) for women. For men, there was a negative relationship between DVPAm and "hopeless" (D4).

### Intra-subject network analysis

Figure 2 (i.e., the intra-subject network (top)) shows that the four previously noted clusters in the inter-subject network analysis appeared clearer. The links between the clusters were weaker except for the strongest link between "nervousness" (an8c) and "hopelessness" (D4c). There were fewer connections between the PA frequency dimensions and the other clusters than in the inter-subject network. Only two (DMPAc and DVPAc) were still positively connected with the "attentive" variable (ap10c). The other link worth noting is the negative relationship of DVPAc with "distressed" (an1c).

Figure 2 (middle and button) shows the results in the within-subjects network by participant gender. The four previously noted clusters that emerged in the full sample were present for both men and women. The links between clusters were also weaker, except between "nervous" (an8c) and "hopeless" (D4c) for women and between "irritable" (an2c) and "hopeless" (D4c), between "distressed" (distressed) (an1c) and "unhappy" (D3c), and

**Table 1.** Descriptive analyses of physical activity levels and mental health variables

| | T1 | | | | | | T2 | | | | | |
| | Women | | Men | | Overall | | Women | | Men | | Overall | |
| | M | SD | M | SD | M | SD | M | SD | M | SD | M | SD |
| Tired – D1 | 1.08 | 1.02 | .94 | .95 | 1.02 | .99 | 1.28 | .94 | 1.13 | .90 | 1.21*** | .92 |
| Sleep – D2 | 1.22** | 1.02 | .96 | .99 | 1.11 | 1.02 | 1.51* | 1.01 | 1.32 | .97 | 1.43*** | 1.00 |
| Unhappy – D3 | 1.26*** | .99 | .89 | .90 | 1.10 | .97 | 1.38*** | .89 | .99 | .88 | 1.21* | .90 |
| Hopeless – D4 | 1.19*** | .99 | .84 | .92 | 1.04 | .98 | 1.33*** | .93 | .99 | .92 | 1.18* | .94 |
| Tense – D5 | 1.40*** | .96 | 1.07 | .90 | 1.26 | .95 | 1.45*** | .87 | 1.10 | .88 | 1.30 | .89 |
| Worried – D6 | 1.51*** | .94 | 1.14 | .91 | 1.35 | .94 | 1.53*** | .90 | 1.20 | .87 | 1.39 | .90 |
| Interested – ap1 | 2.79 | 1.05 | 2.85 | .92 | 2.82 | 1.00 | 2.73 | 1.02 | 2.83 | .89 | 2.77 | .96 |
| Excited – ap2 | 1.46 | 1.20 | 1.70* | 1.10 | 1.56 | 1.16 | 1.59 | 1.17 | 1.85** | 1.11 | 1.70* | 1.15 |
| Strong – ap3 | 2.47 | .99 | 2.51 | .94 | 2.49 | .97 | 2.45 | 1.03 | 2.56 | .98 | 2.50 | 1.01 |
| Enthusiastic – ap4 | .95 | 1.14 | 1.21** | 1.10 | 1.06 | 1.13 | 1.16 | 1.21 | 1.46** | 1.10 | 1.28** | 1.17 |
| Proud – ap5 | 2.20 | 1.11 | 2.34 | 1.03 | 2.26 | 1.08 | 2.28 | 1.09 | 2.40 | 1.07 | 2.33 | 1.08 |
| Alert – ap6 | 2.57 | 1.10 | 2.48 | 1.05 | 2.53 | 1.08 | 2.42 | 1.11 | 2.50 | 1.03 | 2.46 | 1.08 |
| Inspired – ap7 | 1.56 | 1.17 | 1.74 | 1.09 | 1.64 | 1.14 | 1.61 | 1.19 | 1.81* | 1.11 | 1.69 | 1.16 |
| Determined – ap8 | 2.05 | 1.15 | 2.16 | 1.05 | 2.10 | 1.11 | 2.01 | 1.14 | 2.28** | 1.09 | 2.13 | 1.12 |
| Active – ap9 | 2.00 | 1.13 | 2.06 | 1.07 | 2.02 | 1.10 | 1.98 | 1.12 | 2.16* | 1.09 | 2.06 | 1.11 |
| Attentive – ap10 | 2.37 | 1.09 | 2.26 | 1.07 | 2.32 | 1.08 | 2.28 | 1.14 | 2.39 | 1.13 | 2.33 | 1.14 |
| Distressed – an1 | 1.19*** | 1.14 | .86 | 1.01 | 1.04 | 1.10 | 1.08** | 1.08 | .83 | .97 | .97 | 1.04 |
| Irritable – an2 | 1.68*** | 1.15 | 1.38 | .96 | 1.55 | 1.08 | 1.57** | 1.11 | 1.33 | 1.01 | 1.47 | 1.07 |
| Guilty – an3 | .30 | .66 | .25 | .70 | .28 | .68 | .25 | .59 | .23 | .59 | .24 | .59 |
| Scared – an4 | 1.33*** | 1.06 | .72 | .82 | 1.07 | 1.01 | 1.15*** | 1.04 | .72 | .85 | .96 | .99 |
| Hostile – an5 | .28 | .65 | .36 | .66 | .31 | .66 | .33 | .71 | .48 | .81 | .40* | .76 |
| Irritable – an6 | .84 | 1.05 | .91 | .99 | .87 | 1.03 | .83 | 1.02 | .89 | 1.03 | .86 | 1.02 |
| Ashamed – an7 | .14 | .48 | .24 | .67 | .18 | .57 | .20 | .56 | .26 | .65 | .23 | .60 |
| Nervous – an8 | 1.19* | 1.17 | .95 | 1.05 | 1.08 | 1.12 | 1.12* | 1.08 | .92 | 1.00 | 1.04 | 1.05 |
| Jittered – an9 | 1.01* | 1.17 | .80 | 1.00 | .92 | 1.10 | .98 | 1.07 | .88 | 1.02 | .94 | 1.05 |
| Afraid – an10 | 1.09*** | 1.20 | .60 | .87 | .88 | 1.10 | .98*** | 1.08 | .56 | .87 | .80 | 1.02 |
| DVPA | 1.52 | 2.09 | 1.89* | 2.26 | 1.68 | 2.17 | 1.72 | 2.27 | 2.26** | 2.42 | 1.95* | 2.35 |
| DMPA | 2.51 | 2.40 | 2.52 | 2.43 | 2.51 | 2.41 | 2.88 | 2.52 | 2.92 | 2.57 | 2.90** | 2.54 |
| DLPA | 4.67* | 2.52 | 4.20 | 2.68 | 4.47 | 2.60 | 4.94** | 2.46 | 4.34 | 2.74 | 4.68 | 2.60 |

*Note*: * $p < .05$, ** $p < .01$, and *** $p < .001$. DVPA, DMPA, and DLPA represent days per week of vigorous, moderate, and light physical activity, respectively.

between "excited" (excited) (ap2c) and "distressed" (distressed) (an1c) for men. For women, DLPA was not connected with any variable, while in men it was negatively related to "irritable" (an2c) and "guilty" (an3) and positively related to "sleep" (D2c) and "unhappiness" (D3). In the DMPAc, the only connection worth mentioning was the positive connection seen in women with "attentive" (ap10c), while in the DVPAc, the most important difference was the positive connection with "inspired" (ap2c), which was not apparent for men.

## Network indexes

The CS coefficients obtained in both inter- and intra-subject network analyses were above 0.74, indicating a proper value.

Figure 3 shows the centrality indices of all estimated nodes in the inter-subject and intra-subject networks for the whole sample and men and women separately. In the inter-subject network, the "unhappiness" (D3m) node showed the highest centrality of betweenness, closeness, and strength for both the total sample and men. There was a similar pattern for women except for the centrality of intermediation. In this case, the highest variable was "tiredness" (D1m). More differences were seen in the intra-subject network. In terms of closeness centrality, "embarrassed" (an7c) was the strongest variable connected to other nodes in the shortest path, both for the total sample and for women, while in men the strongest was "unhappiness" (D3m). The strength centrality data indicate that "inspired" (ap7c) was the most central node for both the total sample and for women, while in men the strongest was

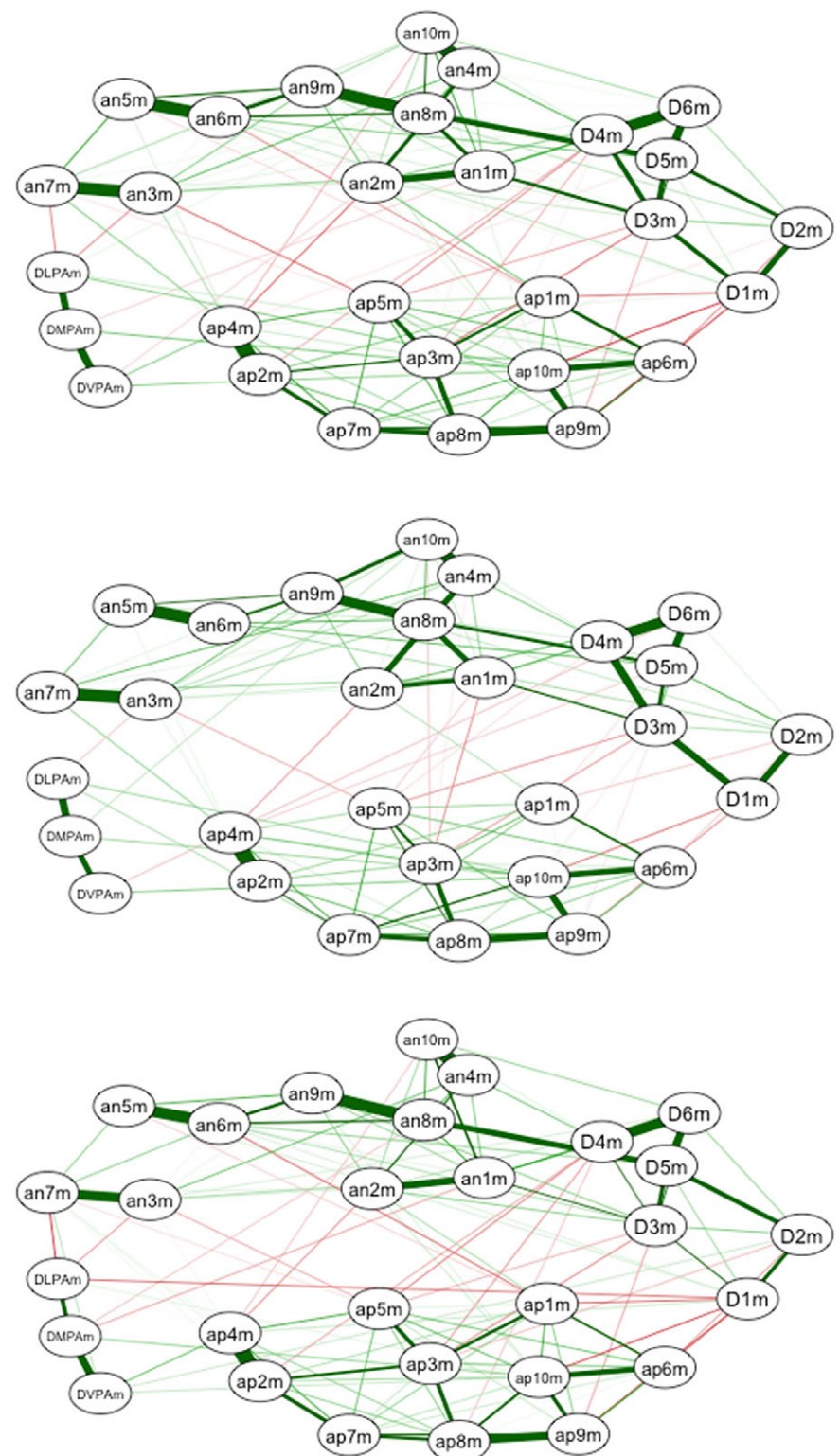

**Figure 1.** Inter-subject network analysis for the full sample (top), men (middle), and women (bottom). *Note.* Men = above, women = below.

"determined" (ap8c). Analyses of betweenness centrality identified the symptom of "unhappiness" (D3m) as the node most frequently found on the shortest path between other nodes for both the full sample and for men, while in women it was "embarrassed" (an7c).

### Discussion

To our knowledge, this is the first study that has adopted a network approach to examine associations between PA, depression, positive affect, and negative affect at two time points during

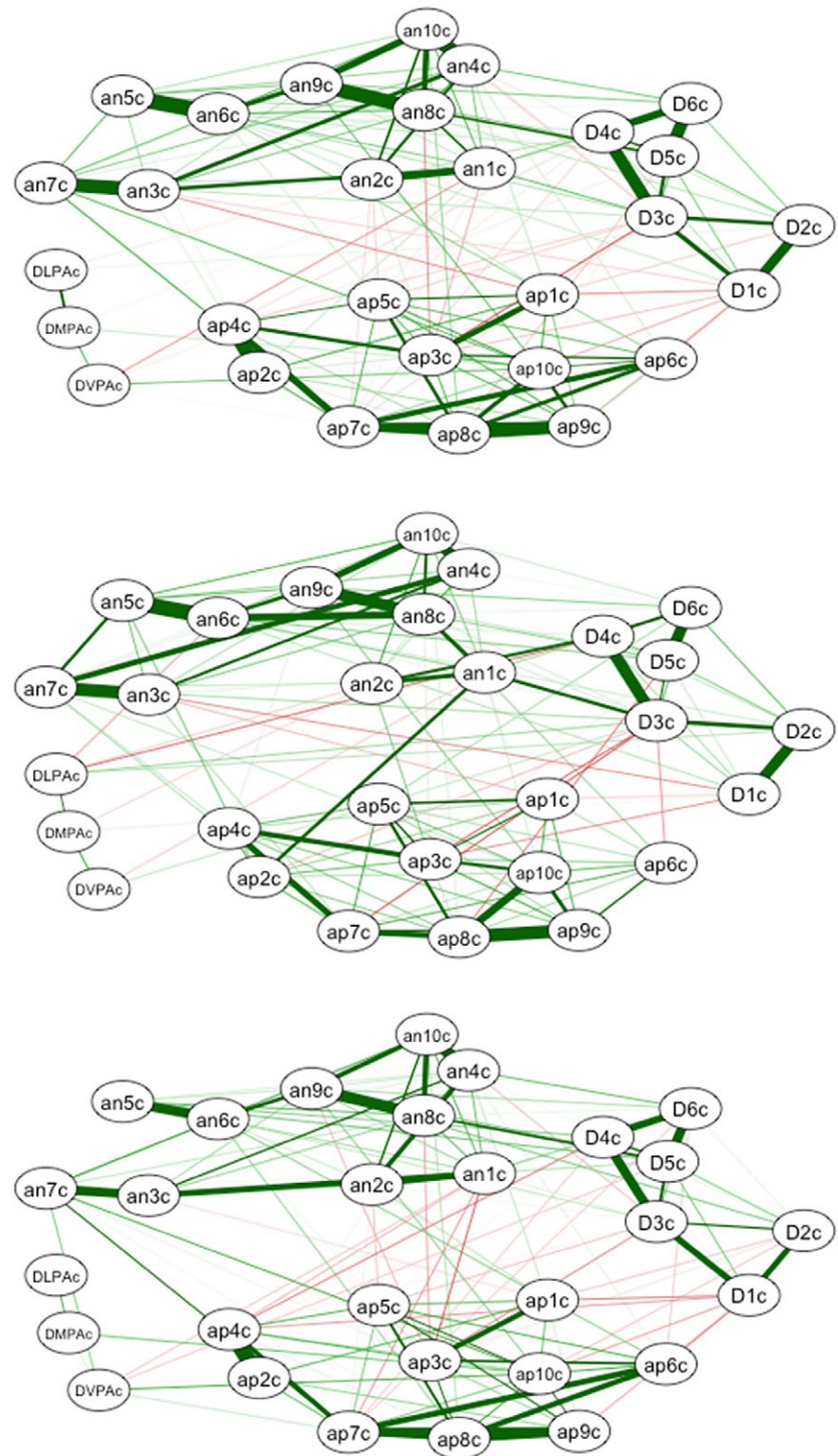

**Figure 2.** Intra-subject network analysis for the full sample (top), men (middle), and women (bottom). *Note.* Men = above, women = below.

the confinement of the population to their homes during the COVID-19 pandemic. The results showed four clusters (PA, depression, positive affect, and negative affect), which were consistent in the within-subject network and the between-subjects network for both men and women. The four clusters were not completely separated, as they were connected with weaker links, except for the strong links between nervousness and hopelessness and between distress and unhappiness. The isolation of people in their homes by COVID-19 seems to have increased the state of nervousness, which was related to a feeling of hopelessness for the future. In addition, the anguish of feeling confined led to a feeling of sadness and unhappiness, and vice versa.

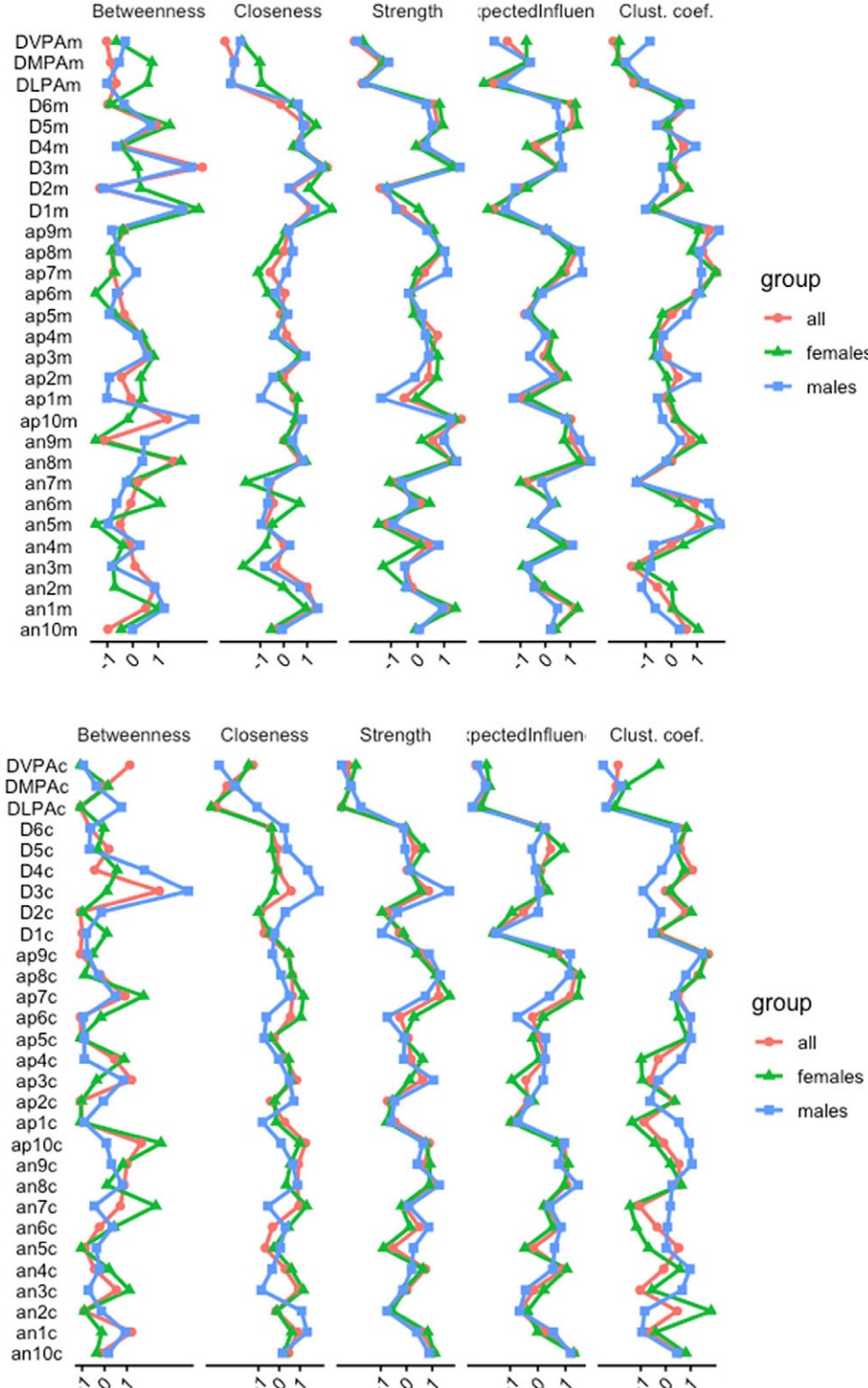

**Figure 3.** Network indexes for intra-subject network analysis and inter-subject network analysis. *Note.* Above = inter-subjects; below = intra-subjects.

Depression nodes play a key role in the inter-subject network, in line with previous studies (Fitzpatrick et al., 2020; Zhang et al., 2020). The sadness and unhappiness symptom node showed the highest centrality of mediating closeness and strength for both the full sample and for men. For women, we found the same pattern, except for the centrality of betweenness, a parameter space that was occupied by another depression symptom (i.e., feeling too tired to get things done). These results are consistent with findings from Di Blasi et al. (2021), which put depression in a central place, meaning it could have a considerable impact on both positive and negative affect symptoms.

In the within-subjects network, centrality was shared by symptoms of depression and affection. However, the centrality of closeness and intermediation in men was occupied by the same depression symptom as in the inter-subject network – "unhappiness." In women, the centrality of closeness and mediation was occupied by the negative affect node "ashamed." The strength centrality data were occupied by nodes of positive affect, "inspired" (for the total sample and women), and "determined" (men).

In the between-subjects network for the full sample, the connections of the PA dimensions were with the two affect clusters, which were the closest. In particular, there were positive connections with positive affect symptoms and negative connections with negative affect symptoms. In other words, PA during isolation had a positive effect on confined people's emotional states. All three PA dimensions were positively connected with the attentive positive affect variable (attentive). This may be because PA is related to neuroelectrical activity, suggesting better overall sustained attention, which demonstrates a better ability to allocate attentional resources over time (Luque-Casado et al., 2016). Feelings of guilt and shame seemed to be able to be mitigated by the frequency of light PA, which also seemed to positively influence alertness. The frequency of moderate PA seemed to reduce distress, while the frequency of vigorous PA decreased irritability and increased enthusiasm. Ultimately, the frequency of all three PA dimensions exhibited beneficial effects on confined people's emotions.

In the within-subjects network, there were fewer connections of the PA frequency dimensions with the other clusters than in the between-subjects network. Only DMPAc and DVPAc were still positively connected with the "attentive" symptom. The other link worth noting was the negative relationship of DVPAc with "distressed."

In the inter-subject network, direct relationships between PA and depressive symptoms only appeared in women, with a negative connection between PLDm and feeling too tired to do things. In men, there was a negative connection between PWDm and feeling hopeless about the future. In the within-subjects network, curiously, there was a positive relationship in men between PWDm and having problems falling asleep and feeling unhappy.

Stress, such as social isolation, is an environmental factor that precipitates a potent physiological response involving the autonomic nervous system and the hypothalamic–pituitary–adrenal axis (Lee, 2022). The brain is especially susceptible to the catabolic effects of stress and its glucocorticoid hormone cortisol (GC), as prolonged exposure to stress or cortisol leads to the development of psychiatric disorders such as anxiety and depression. Social isolation further affects synaptic plasticity and increases basal synaptic transmission in hippocampal CA1 pyramidal neurons, and it appears that physical exercise prevents stress-induced synaptic effects (Aberg et al., 2012).

A recent systematic review examining the association between physical activity and mental health during the first year of the COVID-19 pandemic (Marconcin et al., 2022) reported that greater PA is associated with greater well-being, quality of life, and lower depressive symptoms, anxiety, and stress, regardless of age. Sang et al. (2021) tested a structural equation model to examine the psychological impacts of COVID-19 home confinement and PA, indicating that PA is the best strategy to manage psychological issues such as depression and anxiety. In the present study, PA was found to be associated with better-sustained attention, increased alertness, and enthusiasm. In addition, it mitigates feelings of guilt and shame and decreases confinement distress and irritability. In women, it also reduced fatigue, and in men, it reduced feelings of unhappiness. To our knowledge, this is the first study to address the relationship between PA and depressive and affective symptoms, so it is not possible to compare results with previous research. Nevertheless, the results support the need to increase PA during confinement. However, studies that collected measures before and after the stay-at-home order found a significant reduction in PA (Savage et al., 2020; Suzuki et al., 2020), and depressive symptoms increased as weeks of isolation passed (Cecchini et al., 2021). In addition, increased levels of physical activity have been associated with stronger effects on well-being (O'Brien and Forster, 2021), with a decrease in depressive symptoms (Marconcin et al., 2022) and an increase in positive affect (Carriedo et al., 2020b).

There are several research studies that have analyzed the relationships between PA and different psychological factors during home confinement (e.g., López-Bueno et al., 2020; Carriedo et al., 2020a, 2020b; Cecchini et al., 2021; Dai et al., 2021; Méndez-Giménez et al., 2021; Sang et al., 2021 and Fernandez-Rio et al., 2022). However, the results of this study are novel because it uses a new methodology, which is complementary to other types of study, to understand the relationship between depression, positive and negative affects, and physical activity, where symptoms are secondary to a common underlying cause (Borsboom, 2008; Schmittmann et al., 2013).

PA was an effective option for mitigating the negative effects of the COVID-19 pandemic on mental health during its first year. Public health policymakers should be alert to possibilities of increasing physical activity during stay-at-home orders in many countries around the world. Moreover, programs should be developed to promote physical activity to be able to cope with confinement or similar scenarios in the future. For instance, programs that describe recommendations and guidelines for staying active at home, with aerobic exercise training on a bike or rowing ergometer, bodyweight training, dance, and videos with full-body active routines, can be useful to achieve the recommended physical activity levels that might mitigate the negative effects of the COVID-19 pandemic. Also, fitness apps, live-streaming workout classes, and virtual reality fitness for PA could be useful (Liu et al., 2022).

This study has some limitations. For instance, this methodological approach might lead to the overfitting of the model. For this reason, although network approaches are a useful exploratory tool, the presented results need to be confirmed in future research (cross-validation or experimentation; Epskamp et al., 2018). Nevertheless, this network approach can be a very useful tool for modeling different dynamic relationships between psychological constructs addressing emotional, psychological, and social well-being and PA, for example, affective-type disorders, or personality and behavioral disorders, psychological developmental disturbances, and their relationship with PA. Another limitation is related to the sample criteria. Depressive symptoms and affective symptoms could be affected by one's medical conditions. In this study, this information had not been collected; therefore, this should be also considered in future studies.

**Open peer review.** To view the open peer review materials for this article, please visit http://doi.org/10.1017/gmh.2023.57.

**Data availability statement.** Data are available for research. Any further inquiries can be directed to the authors.

**Author contribution.** J.A.C., A.C., A.M.-G., and J.F-R conceptualized the study; J.A.C. designed methodology; A.C. provided software; A.M.-G. and J.F-R. validated the study; J.A.C. involved in formal analysis; A.M.-G. and J-F-R

investigated the study; A.C. curated the data; J.A.C. wrote the original draft preparation; A.C., A-M-G., and J- F-R wrote, reviewed, and edited the manuscript; and all authors have read and agreed to the published version of the manuscript.

**Competing interest.** The authors report no conflicts of interest.

**Ethics standard.** Permission to conduct the study was obtained from the University of Oviedo Research Ethics Committee (no. 2020.165).

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
