## [Reviewer Report]

Dear Editor:

Please find enclosed a manuscript entitled: “Network analysis of physical activity and depressive and affective symptoms during Covid-19 home confinement” which I am submitting for exclusive consideration of publication as an article in CAMBRIDGE PRSMS: GLOBAL MENTAL HEALTH.

The corresponding authors of this manuscript are José A. Cecchini, Alejandro Carriedo, Antonio Méndez Giménez, and Javier Fernández Río from the University of Oviedo, and contribution of the authors as mentioned below with their responsibility in the research is to analyze the network structure of physical activity (PA) frequency, depressive, and affective symptoms in people under home isolation due to COVID-19. This is the first study that have analysed these variables during the confinement through the Network Analysis approach.

All authors of this research paper have directly participated in the planning, execution, or analysis of this study;

• The research ethics committee of the University of Oviedo reviewed and approved this study

• All authors of this paper have read and approved the final version submitted;

• All authors reveal no conflict of interest in the conduct and reporting this research;

• The contents of this manuscript have not been copyrighted or published previously;

• The contents of this manuscript are not now under consideration for publication elsewhere;

• The contents of this manuscript will not be copyrighted, submitted, or published elsewhere, while acceptance by the Journal is under consideration;

• There are no directly related manuscripts or abstracts, published or unpublished, by any authors of this paper;

• The University of Oviedo is fully aware of this submission;

• Submitted manuscript is an original research;

• Alejandro Carriedo is the corresponding author (carriedoalejandro@uniovi.es);

Thank you for your consideration of our work 

Sincerely

Dr. Alejandro Carriedo

---

## [Reviewer Report]

This manuscript, “Network analysis of physical activity and depressive and affective symptoms during Covid-19 home confinement,” aims at investigating the network structure of physical activity (PA) frequency and depressive and affective symptoms in people under home isolation due to COVID-19. The authors used the IPAQ short form (IPAQ-SF) to pursue this aim. Despite several major issues detailed below, the authors sufficiently fulfill their aim.

1. Introduction section is organized poorly without logic and sequence. Authors should organize the introduction section with subheadings. The following recommended articles can help reorganize the introduction section;

https://doi.org/10.3389/fpubh.2022.852311

https://doi.org/10.2147/JMDH.S354984

https://doi.org/10.2147/PRBM.S350666

https://doi.org/10.3389/fpsyg.2021.667461

2. Secondly, remove all acronyms from the abstract.

3. Thirdly, please mention the time period and place for data collection.

4. Why does network analysis explore associations among different activities/concepts such as PA, depressive symptoms and positive and negative effects? For this purpose, the regression analysis, such as the structural regression method known as SEM, should be included to examine the positive and negative associations. The following papers can be helpful in the revision of the statistical analysis

https://doi.org/10.3389/fpsyg.2020.614770

5. For a better understanding of the aim, research questions should be proposed in the introduction chapter

6. The status of elected concepts is not explained well. There have been only 4 actions happening during covid period in Spain.

7. The theoretical linkage for relations among elected construct is mislaid.

8. How about the controls such as income, job, education, and age of respected participants to explore the relations among elected constructs?

9. What was the reason for excluding 62 participants during the analysis?

10. There were two waves for data collection, right? In the second wave, all participants were excluded; why?

11. How about the reliability and validity test for the data collection instrument? Before analyzing the relations among constructs, there should be a reliability and validity examination for the final questionnaire.

12. Regarding the results obtained analysis, I think the paper could be stronger if a closer link with previous studies on related topics is included.

13. Managerial and practical implications are missing in the discussion chapter.

14. The final section (discussion and conclusion) is poor and should be revised with latest literature. Following are the latest articles which could be helpful

https://doi.org/10.3389/fpubh.2022.852311

https://doi.org/10.2147/JMDH.S354984

https://doi.org/10.2147/PRBM.S350666

https://doi.org/10.3389/fpsyg.2021.667461

https://doi.org/10.3389/fpsyg.2020.614770

https://doi.org/10.2147/PRBM.S369020

---

## [Reviewer Report]

? I thank the authors’ efforts in examining the network structure of PA frequency, and depressive and affective symptoms in people during COVID-19 home confinement. Even though the topic is interesting, there are several major concerns that need to be addressed. First, the study said “a new approach to understanding the complex relationships between psychiatric symptoms has emerged, called network analysis” but this has been out there for a long time. Here, it is still unclear how the previous relevant studies that the authors reviewed (e.g., Carriedo et al., 2020a, 2020b, Fernandez-Rio et al., 2022, Lopez-Bueno et al., 2020, Cecchini et al., 2021). A more direct and clear comparison should be provided in the introduction. Also, the author argued “no study so far has used network analysis to explain the relationship of PA with depressive symptoms and positive and negative affect during home isolation”. Nevertheless, several studies examined the relationship between PA, depressive symptoms, and affective symptoms. Even if these studies didn’t use network analysis, the authors should clearly explain the differences between these previous studies and the present study. Please review those latest articles more thoroughly. Related to this, the introduction section should be enhanced further. More clear problem statement should be provided as well as more literature background on relationships among PA, depressive symptoms, and affective symptoms among the specific research populations.

2. There are several major issues in methods as well. The participants should be explained in more detail – recruitment methods, recruitment procedures (contacting email, facebook, whatsapp, and twitter is not enough. How did you choose potential participants to contact via these methods?), population boundary, sample boundary, and sample selection criteria. Especially, depressive symptoms and affective symptoms will be affected by one’s personal conditions such as medical conditions, it is important to set the inclusion/exclusion criteria clearly to control those extraneous variables. More information should be included. Also, when it comes to network analysis, several critical elements were missing – e.g., the definition of nodes, ties, modes of the network, network boundary, etc.

3. Since the critical information was missing, it’s hard to assess if the findings were properly laid out. More importantly, the rationale for this study was that this study used network analysis. Nevertheless, it’s unclear how the results of this study can be compared to the other relevant previous studies in the discussion section. More in-depth discussion should be added as well as original theoretical, methodological, and practical implications.

4. Lastly, there are too many awkward sentences – I highly encourage the author(s) to proofread the manuscript more thoroughly to avoid grammatical errors and improper sentence structures.

[detailed comments]

1. Abstract – “method” -> “methods”

2. Abstract – in methods, what kinds of network analysis were performed? Measurement?

3. P. 2, line “… is warning about”: should be “has been warning” or “has warned”

4. Throughout the document, the author(s) implies that we are in the middle of the COVID-19 pandemic. Yet, the time has passed and there have been many updates in terms of academic literature, policy, and so forth. Hence, the authors should revise the whole manuscript accordingly.

5. I’d use “betweenness centrality” rather than intermediation centrality

---

## [Reviewer Report]

Dear Editor:

Please find enclosed a manuscript entitled: “Network analysis of physical activity and depressive and affective symptoms during Covid-19 home confinement” which I am submitting for exclusive consideration of publication as an article CAMBRIDGE PRISM: GLOBAL MENTAL HEALTH.

The corresponding authors of this manuscript are José A. Cecchini, Alejandro Carriedo, Antonio Méndez Giménez, and Javier Fernández Río from the University of Oviedo, and contribution of the authors as mentioned below with their responsibility in the research is to analyze the network structure of physical activity (PA) frequency, depressive, and affective symptoms in people under home isolation due to COVID-19. This is the first study that have analysed these variables during the confinement through the Network Analysis approach.

All authors of this research paper have directly participated in the planning, execution, or analysis of this study;

• The research ethics committee of the University of Oviedo reviewed and approved this study

• All authors of this paper have read and approved the final version submitted;

• All authors reveal no conflict of interest in the conduct and reporting this research;

• The contents of this manuscript have not been copyrighted or published previously;

• The contents of this manuscript are not now under consideration for publication elsewhere;

• The contents of this manuscript will not be copyrighted, submitted, or published elsewhere, while acceptance by the Journal is under consideration;

• There are no directly related manuscripts or abstracts, published or unpublished, by any authors of this paper;

• The University of Oviedo is fully aware of this submission;

• Submitted manuscript is an original research;

• Alejandro Carriedo is the corresponding author (carriedoalejandro@uniovi.es);

Thank you for your consideration of our work 

Sincerely

Dr. Alejandro Carriedo